# C1-FDX is required for the assembly of mitochondrial complex I and subcomplexes of complex V in Arabidopsis

**Baoyin Chen**[1,2]ʘ, **Junjun Wang**[1]ʘ, **Manna Huang**[1], **Yuanye Gui**[1], **Qingqing Wei**[1], **Le Wang**[1], **Bao-Cai Tan**[1]*

**1** Key Laboratory of Plant Development and Environmental Adaptation Biology, Ministry of Education, School of Life Sciences, Shandong University, Qingdao, China, **2** College of Agriculture, and State Key Laboratory of Crop Biology, Shangdong Agricultural University, Tai'an, China

ʘ These authors contributed equally to this work.
* bctan@sdu.edu.cn

## Abstract

C1-FDX (Complex I-ferredoxin) has been defined as a component of CI in a ferredoxin bridge in Arabidopsis mitochondria. However, its full function remains to be addressed. We created two *c1-fdx* mutants in Arabidopsis using the CRISPR-Cas9 methodology. The mutants show delayed seed germination. Over-expression of C1-FDX rescues the phenotype. Molecular analyses showed that loss of the C1-FDX function decreases the abundance and activity of both CI and subcomplexes of CV. In contrast, the over-expression of C1-FDX-GFP enhances the CI* (a sub-complex of CI) and CV assembly. Immunodetection reveals that the stoichiometric ratio of the α:β subunits in the F1 module of CV is altered in the *c1-fdx* mutant. In the complemented mutants, C1-FDX-GFP was found to be associated with the F' and α/β sub-complexes of CV. Protein interaction assays showed that C1-FDX could interact with the β, γ, δ, and ε subunits of the F1 module, indicating that C1-FDX, a structural component of CI, also functions as an assembly factor in the assembly of F' and α/β sub-complexes of CV. These results reveal a new role of C1-FDX in the CI and CV assembly and seed germination in Arabidopsis.

**Data Availability Statement:** All relevant data are within the manuscript and its Supporting Information files.

## Author summary

The mitochondrial oxidative phosphorylation (OXPHOS) system is composed of five large multiprotein complexes, complex I to V (CI to CV). However, their assembly is not fully understood in plants. C1-FDX, a ferredoxin protein, was found to form a ferredoxin bridge in CI. Here, we show that C1-FDX is also involved in the assembly of CI* and CV. CI* is an important intermediate of CI. Mutation of C1-FDX affects the assembly of these two complexes and delays the germination of Arabidopsis. C1-FDX interacts with several CV subunits, including the β, γ, δ, and ε subunits of the F1 sub-complex, and physically associates with the CV assembly intermediates, sub-complexes F' and α/β, but not with the mature CV, indicating that C1-FDX is dissociated when the CV is fully assembled.

**Funding:** This research was supported by the National Natural Science Foundation of China (www.nsfc.gov.cn/) (32230075 to BCT and 32001610 to BC). The funders had no role in study design, data collection and analysis, decision to publish, or preparation of the manuscript.

**Competing interests:** The authors have declared that no competing interests exist.

This study provides insights into the dual-function of CI-FDX in CI and CV assembly, facilitating the understanding of the CI and CV assembly mechanism in plants.

## Introduction

Mitochondria, as the powerhouse of the cell, synthesize ATP through an oxidative phosphorylation system (OXPHOS) located in the inner membrane of mitochondria. OXPHOS comprises five large protein complexes, named complex I to V (CI-CV). NADH is oxidized by CI and $FADH_2$ by CII (succinate dehydrogenase) to produce two electrons sequentially transferred through CIII and CIV. During the electron transfer, protons are pumped by CI, CIII, and CIV from inside the inner membrane to the intermembrane space, generating a proton gradient across the inner membrane. Protons transferring back through CV (also called ATP synthase or F1Fo ATP synthase) drive the synthesis of ATP [1–3]. In eukaryotes, several core subunits of CI, CII, and CIII are iron-sulfur (Fe-S) proteins that play a key role in the electron transfer chain [4]. The expression of these proteins assembled with the iron-sulfur (Fe-S) cluster is essential to the assembly and activity of the OXPHOS complexes.

Ferredoxins (FDXs) are redox proteins containing the iron-sulfur [2Fe-2S] cluster. Because iron atoms can transit between ferrous (+2) and ferric (+3) states, ferredoxins can function as electron transfer agents in a range of redox reactions and also as an assembly factor for the assembly of Fe-S cluster proteins [5,6]. The first FDX was identified in plant chloroplast, which is involved in photosystem I (PSI) of photosynthesis [6]. In contrast, less attention has been given to the mitochondrial FDXs, especially in plants. Three FDXs are identified in Arabidopsis and rice mitochondria, named mitochondrial ferredoxin 1 (mFDX1) or Adrenodoxin 2 (ADX2), mitochondrial ferredoxin 2 (mFDX2) or Adrenodoxin 1 (ADX1), and C1-FDX, respectively [7]. The Arabidopsis mFDX1 and mFDX2 are probably paralogs with 87% amino acid sequence identity [4], and both contain four conserved cysteine (Cys) residues in the core domain binding loop [2]. In humans, mFDX1 and mFDX2 play an essential role in the assembly of the Fe-S cluster and the biosynthesis of heme A [5]. In Arabidopsis, mFDX1 and mFDX2 are crucial for maternal gametophytic control of embryogenesis, which is associated with steroid hormone synthesis and biotin synthesis [8,9]. Compared with Arabidopsis mFDX1 and mFDX2, the Arabidopsis C1-FDX lacks a conserved Cys residue and shares merely 21% amino acid sequence identity with the two mFDXs [4]. Recently, C1-FDX has been found in CI forming a ferredoxin bridge domain connecting the ubiquinone reduction module (Q module) of the matrix arm to the γ-carbonic anhydrase domain (CA domain) of the membrane arm in Arabidopsis, green alga *Polytomella*, *Vigna radiata* and *Tetrahymena thermophila* [2,10–12]. C1-FDX is essential for forming the supercomplex CI+III₂ [13]. The ferredoxin bridge consists of C1-FDX, A6 (also called B14 or NB4M), and ACP2 (also named SDAP2, ACPM1) of the CI subunits [2]. C1-FDX and the ferredoxin bridge were not found in yeast *Yarrowia lipolytica*, but A6 and ACP2 are found to be attached to the Q module and critical for CI activity [14], implying that C1-FDX may have other functions in plant mitochondria.

Plant CI is a multimeric NADH:ubiquinone oxidoreductase consisting of more than 48 subunits, which composes the $P_P$ (proximal proton-pump), $P_D$ (distal proton-pump), N (NADH oxidation), and the Q and CA modules mentioned above [15]. CI is assembled modularly, where each module assembles first, then the modules assemble into the holo-CI [3,15]. Thus, a deficiency of any module arrests the assembly at specific stages and blocks the production of CI [15–18]. For example, a deficiency of the $P_D$ module stops CI assembly at the CI*

sub-complex stage, whereas a deficiency of the Q or N module leads to the formation of only the membrane arm [15,16]. The ferredoxin bridge is a newly discovered module [2,12], and its role in CI assembly is unknown.

CV is also a multi-subunit complex with an F1 and an Fo module [1]. The F1 module located in the mitochondrial matrix catalyzes the synthesis of ATP, and the Fo module located in the inner membrane serves as a proton-pumping channel. The F1 module consists of nine subunits, α(3), β(3), γ(1), δ(1), and ε(1). Three α and β subunits form the $\alpha_3\beta_3$ hexamer catalytic head, and the γ, δ, and ε subunits form the central stalk [1]. The Fo module is composed of 6–14 subunits varying in species, forming the motor (a and c) and peripheral stalk (e.g., b, d, h, f, etc.). Similar to CI, the assembly of mitochondrial CV has also been proposed to follow a modular pathway [19,20], but certain differences exist among Arabidopsis, yeast, and humans. In Arabidopsis, the F1 module is first assembled, then inserted into the c-ring and peripheral stalk, and finally completes assembly with the OSCP (oligomycin sensitivity-conferring protein) subunit [20]. In yeast, however, the F1 module first assembles with the peripheral arm to form the sub-complex F1-peripheral arm, then inserts the c-ring and a/8 subunits, forming the complete CV [21]. These two assembly pathways coexist in humans [22]. In the assembly process of the F1 module, the $\alpha_3\beta_3$ hexamer catalytic head needs to form the exactly heterogeneous αβαβαβ. In yeast, CV assembly factors ATP11 and ATP12 were shown to bind to β and α, respectively [23–25]. This binding is proposed to prevent formation of the insoluble α-α and β-β homodimers by forming the soluble ATP12-α and ATP11-β heterodimers [26], while the assembly of the $\alpha_3\beta_3$ hexamer in plants is unknown. In addition, a sub-complex F' was identified by CV in-gel activity analysis in maize [27,28], but its composition and role in the assembly process remain to be determined.

In this study, we investigated the biological function of C1-FDX in Arabidopsis. We show that a loss of the C1-FDX function delays seed germination, reduces the activity and abundance of CI and CV, and interestingly alters the stoichiometric ratio between the α and β subunits in CV. Over-expression of C1-FDX-GFP improves the assembly of CI and CV and its intermediates CI* and F1 or α/β, respectively. Importantly, the C1-FDX exists in the F' and α/β sub-complexes and interacts with four F1 subunits β, γ, δ, and ε. In consequence, the *c1-fdx* mutant contains a substantially decreased ATP level. These results indicate that C1-FDX is not only required for the assembly of CI but also is an assembly factor of CV during seed germination in Arabidopsis mitochondria.

## Results

### The *c1-fdx* mutants show delayed seed germination

The Arabidopsis *C1-FDX* (At3g07480) is an intronless gene encoding a protein with 159 amino acid residues (Fig 1A). To investigate the function of C1-FDX, we created two *C1-FDX* mutants using the CRISPR-Cas9 methodology, *c1-fdx-1* and *c1-fdx-2*. *c1-fdx-1* and *c1-fdx-2* have a single base "T" or "G" insertion at +338bp in the coding region, respectively, causing a frameshift and a translation stop codon next to the insertion (Fig 1A and 1B). Sequencing of the *c1-fdx* transcripts in the mutants confirmed the "T" and "G" insertion (S1A and S1B Fig). This mutation potentially truncates the 46 amino acid sequence of the C-terminus of C1-FDX if it is translated.

Both mutants displayed slow germination compared with the wild-type (Fig 1C). Over-expression of C1-FDX complemented the slow germination phenotype in both mutants (Fig 1C), indicating the loss-of-function of *C1-FDX* affects seed germination. The wild-type seeds began germinating at 24 h after stratification, while the *c1-fdx-1* and *c1-fdx-2* seeds started to germinate at 48 h (Fig 1D). At 60 h, 100% of the wild-type seeds germinated, whereas

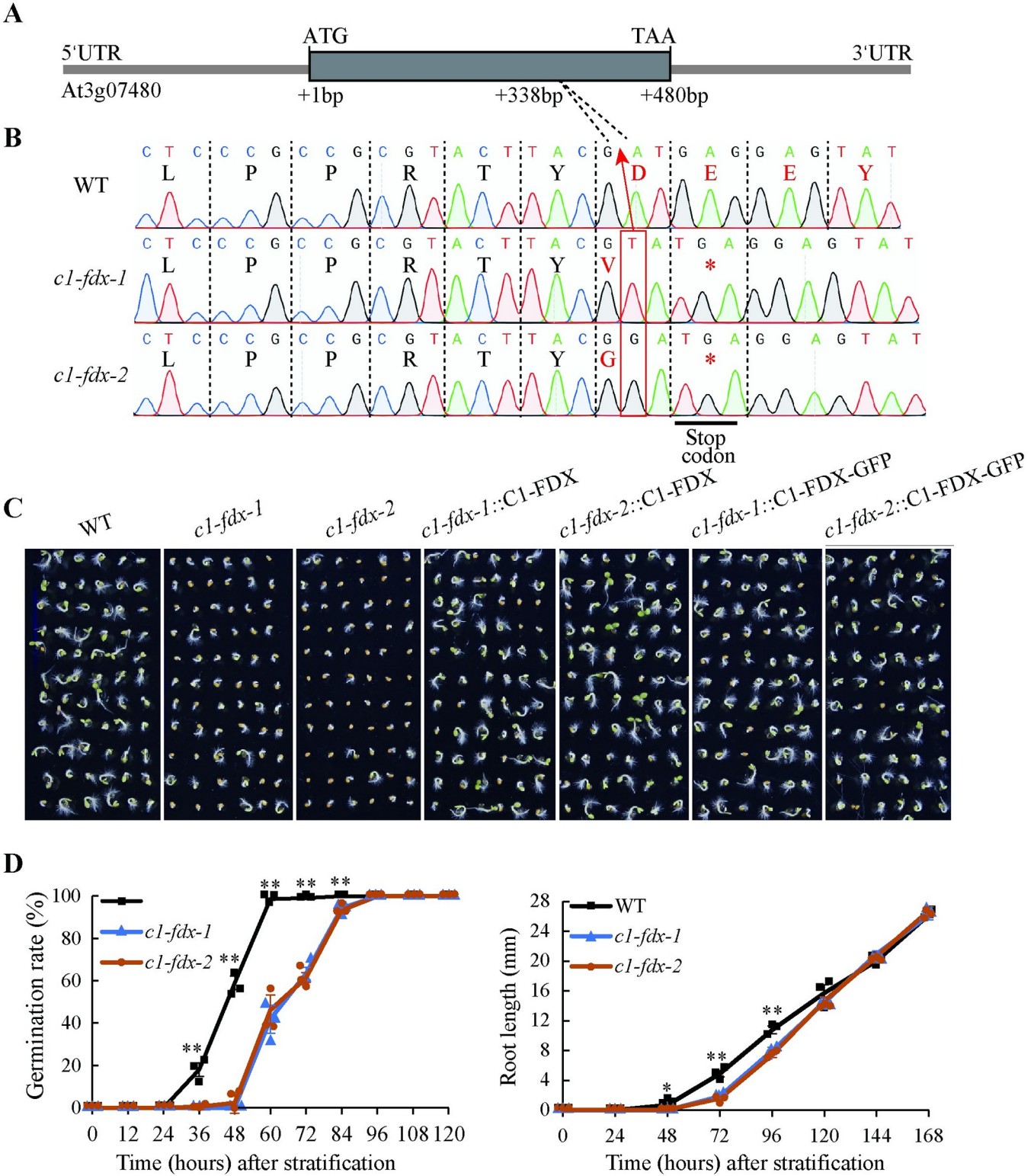

**Fig 1. The *c1-fdx* mutants show delayed seed germination.** (**A**) Gene structure of *C1-FDX* with CRISPR-Cas9 target sites. UTR, Untranslated region. (**B**) Mapping of the CRISPR-Cas9 gene disruption for two individual knockouts of *C1-FDX*. (**C**) The phenotype of *c1-fdx-1* and *c1-fdx-2* growing for 3-day-old *c1-fdx* mutants and its gene-complemented mutants. Complementation of *c1-fdx* mutants with both C1-FDX (expressed by Super promoter) and C1-FDX-GFP (expressed by CaMV 35S) resulted in a restoration of the seed germination delay. (**D**) The germination rate and root length of *c1-fdx-1* and *c1-fdx-2* after stratification. Asterisks indicated significant differences compared with WT plants. **$P<0.01$ and *$P<0.05$, n = 3.

the mutant seeds took 96 h to achieve that (Fig 1D). Slow root growth was also observed in the mutants on the 3rd day of germination under either normal (Fig 1D) or stress conditions (S2 Fig). No significant difference in the cotyledon size and plant height was observed between the wild-type and *c1-fdx* mutants growing for 8 days in MS medium (S3A Fig) and 42 days in soil (S3B Fig). In addition, the *C1-FDX* transcript level in the 3-day-old wild-type seedlings was higher than that in the 8-day-old wild-type seedlings (S3C Fig), indicating that *C1-FDX* plays a role in early seedling development. The expression of *C1-FDX* is barely detectable before 12 h after stratification (S3D Fig). However, the expression of *C1-FDX* was rapidly increased at or after 24 h and decreased at 48 h in the light stage (S3D and S3E Fig), indicating that *C1-FDX* plays a role during Arabidopsis seed germination.

## The activity of CI and CV is reduced in 3-days-old *c1-fdx* seedlings

Previous studies have shown that C1-FDX is a CI component in plant mitochondria [2,10,11,13]. To investigate the function of C1-FDX in seed germination of Arabidopsis, we analyzed the activity of the mitochondrial respiratory chain complexes in *c1-fdx* using the blue native (BN)-PAGE assays. Mitochondrial proteins were isolated from 3-day-old etiolated seedlings cultured in liquid 1/2 MS medium in the dark and resolved by BN-PAGE. The gel was stained with Coomassie brilliant blue (CBB) and analyzed by in-gel enzymatic activity assays (Fig 2A). Compared with the wild-type, the activity of CI and CI* is visibly reduced in *c1-fdx-1* and *c1-fdx-2* (Figs 2A and S4), estimated to be approximately 52% and 58% of the wild-type, respectively (Fig 2B). In addition, the supercomplex (CI+III$_2$) disappeared in *c1-fdx-1* and *c1-fdx-2* when using 4% digitonin solubilization (S4 Fig), indicating that the loss of C1-FDX impedes formation of the CI+III$_2$ supercomplex [13].

Interestingly, the activity of CV and F1 was also reduced in *c1-fdx-1* and *c1-fdx-2* (Figs 2A and S4). The CV and F1 activity in *c1-fdx-1* was significantly reduced by 25% and 50% compared with the wild-type (Fig 2B). To test this reduction, we analyzed the CV activity using the total mitochondrial extract from the 3-days-old etiolated seedlings. The results showed that the CV activity was substantially reduced in the *c1-fdx* mutants (S5 Fig). Oligomycin A is an inhibitor of CV activity, which binds to the Fo subunits a and c to inhibit the CV activity [29]. We treated the wild-type and the *c1-fdx* mutant seedlings with 10 μM Oligomycin A for 3 h. The CV activity in the wild-type and the *c1-fdx* mutants was decreased to a comparable level (S5 Fig). Together, these results indicate that the mutation of C1-FDX reduces both the CI and CV activity. Noted was that the CIV activity was slightly increased in *c1-fdx-1*, and no visible change was observed in the activity of CII and CIII (Fig 2A). Concomitantly, the mitochondrial ATP level was dramatically decreased in the *c1-fdx-1* and *c1-fdx-2* mutants (Fig 2C).

## C1-FDX plays a role in the assembly of CI and CV

To investigate whether the CI and CV activity reduction is caused by defects in the CI or CV assembly in the 3-days-old seedling of the *c1-fdx* mutants, we further analyzed the mitochondrial complexes in Fig 2 by BN-PAGE and immunodetection. Antibodies against subunits of the CI modules (V1 of N module, A5 and Nad9 of Q module, CA2 of P$_P$ module, B10 of P$_D$ module), an assembly factor L-Galactono-1,4-lactone dehydrogenase (GLDH) for detecting early CI complexes, and CV (α, β of the F1 sub-complex, and subunit a of the Fo sub-complex) were used. Of these antibodies, the maize antibodies of V1, A5, Nad9, B10, α recognized its homologous proteins at the expected molecular weight in Arabidopsis (S6 Fig). The rest of the antibodies also recognized the Arabidopsis CI or CV on the BN gel in immunoblotting [15,18,30].

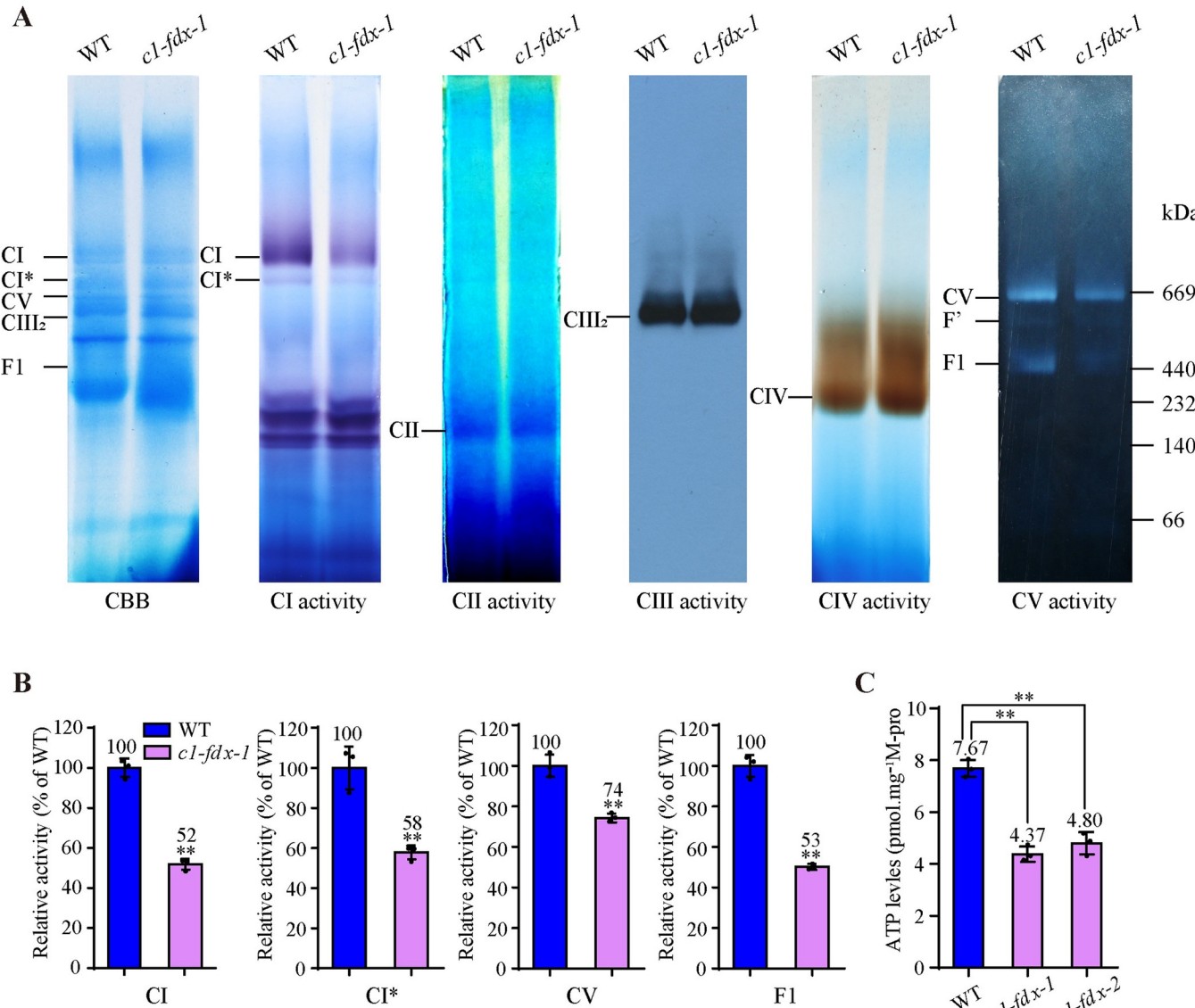

**Fig 2. Loss of C1-FDX results in a reduction of the CI and CV activity. (A)** In-gel activity assays of the CI, CII, CIII, CIV, and CV activity. The mitochondria were solubilized with dodecylmaltoside, and 130 μg mitochondrial proteins from wild-type and *c1-fdx-1* were resolved in 3.5–12% BN-PAGE and assayed for activity (refer to methods). F1 and F' are assembly intermediates of CV. **(B)** The relative activity of CI and CV. The relative activities in each band in panel **(A)** were quantified using the Image J software. Asterisks** indicate a significant difference ($P<0.01$). **(C)** Mitochondrial ATP contents in the WT, *c1-fdx-1*, and *c1-fdx-2* seedlings. Data were means ± SD of three biological replicates. Asterisks** indicate a significant difference ($P<0.01$). M-pro, Mitochondrial proteins.

The result showed that the CI abundance was substantially decreased in the *c1-fdx-1* mutant (Fig 3A), consistent with the reduced CI in-gel activity (Fig 2A). Similarly, the level of the major CI assembly intermediate CI* was also decreased in the mutant, as detected by antibodies A5, Nad9, and CA2 (Fig 3A). In contrast, the abundance of the N module was increased in the *c1-fdx-1* mutant as detected by the V1 antibody (Fig 3A), suggesting that the mutation of C1-FDX leads to an accumulation of the N module of the matrix arm. For the membrane arm, we found that the abundance of the 450 kDa $P_P$ module was increased in the mutant, which was detected by the CA2 and GLDH antibodies (Fig 3A). The $P_D$ module was also slightly accumulated in the *c1-fdx-1* mutant (Fig 3A). These results suggest that the mutation of

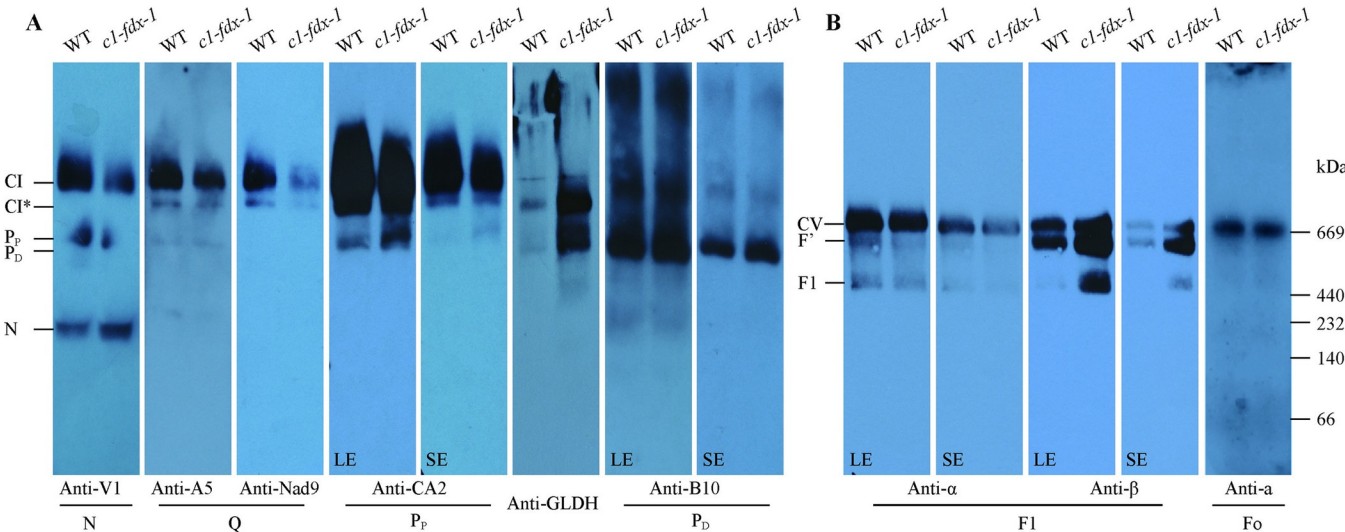

**Fig 3. Loss of the C1-FDX function results in CI and CV assembly defects.** Immunoblotting images of the mitochondrial native complexes separated by BN-PAGE in Fig 2 and hybridized with antibodies against representative subunits from each module of CI (**A**) and CV (**B**). The antibody, the CI (N, Q, $P_P$, $P_D$) and CV modules (F1, Fo), and the corresponding sub-complexes in the gels are labeled. GLDH is a CI assembly factor. F' indicates an assembly intermediate of CV. SE, short-time exposure; LE, long-time exposure.

C1-FDX leads to an accumulation of the unassembled matrix and membrane arm intermediates and a reduced assembly of CI and CI*. GLDH is responsible for preventing the binding of the P1 subunit (a $P_D$ module subunit) to the $P_P$ module, blocking association of $P_D$ with $P_P$ [31]. Interestingly, strong signals of GLDH were detected in CI* and an approximately 450 kDa sub-complex corresponding to the $P_P$ module (Fig 3A), indicating the accumulation of GLDH at the CI* and $P_P$ in the absence of C1-FDX. $P_P$ could assemble with either the matrix arm to form CI* or $P_D$ to form the membrane arm. Incorporation of $P_D$ with CI* forms the complete CI. These results suggest that the mutation of C1-FDX affects the release of GLDH from the $P_P$ or CI* and then blocks the assembly of $P_D$ with $P_P$ or CI*.

CV was hybridized with the α and β antibodies recognizing the $F_1$ subunits and the subunit *a* antibody recognizing one of the Fo module subunits. All three antibodies detected CV (Fig 3B). The α and β antibodies also detected F1 and the recently identified sub-complex F' [27,28]. The abundance of α was decreased in CV, F', and F1 in the *c1-fdx-1* mutant compared with the wild-type. In contrast, the abundance of β was remarkably increased in CV, F', and F1 (Fig 3B). As the α and β subunits are in a strict 1:1 stoichiometric ratio in the $\alpha_3\beta_3$ hexamer [1], the above result indicates that an increased ratio of β in the F1 and F' sub-complexes, and possibly CV too, in the *c1-fdx* mutant. In addition, no change at the position of CV was detected between the wild-type and the *c1-fdx-1* mutant using the subunit *a* antibody (Fig 3B) and Coomassie brilliant blue staining [13] (Fig 3A). The results indicate that the mutation of C1-FDX primarily affects the F1 module assembly.

To further test the impact on the complex subunits, we analyzed the subunits in the total mitochondrial extracts from the 3-days-old seedling by SDS-PAGE and immunodetection. Similar to the results of the BN-PAGE, the abundance of CI subunits V1, A5, Nad9, and CA2 was decreased in *c1-fdx-1*, whereas the CV subunit β and CI assembly factor GLDH were increased (S7 Fig). The abundance of CIII subunit Cyt *c*1 and CIV subunit COX3 showed no visible change in the *c1-fdx-1* mutant (S7 Fig). However, Cyt *c*, a mitochondrial intermembrane space protein that functions as an electron shuttle between CIII and CIV of the respiratory chain, was decreased in the *c1-fdx-1* mutant (S7 Fig). The levels of the AOXs were

increased in *c1-fdx-1* (S8 Fig). These results indicate that the mutation of C1-FDX also affects the abundance of CI and CV subunits and enhances the expression of the alternative pathway, indicator of an impaired oxidative phosphorylation process.

## Over-expression of C1-FDX-GFP improved the assembly of CI and CV

To further test the functions of C1-FDX in CI and CV assembly, we investigated the activity and abundance of CI and CV in the 3-days-old seedling in the C1-FDX-GFP transgenics by BN-PAGE and immunodetection. The results showed that the CI and CI* activity was increased in the *c1-fdx-1*::C1-FDX-GFP complemented mutant (Fig 4B), consistent with an increased abundance of CI and CI* (Fig 4A and 4C), indicating that over-expression of C1-FDX-GFP promotes the assembly of CI and CI*. The fusion protein C1-FDX-GFP was accumulated at the CI position (Fig 4F), confirming that C1-FDX-GFP is functional as a component of CI in Arabidopsis [2].

We found that the abundance of CV, F1, and α/β was increased in the *c1-fdx-1*::C1-FDX-GFP complemented mutant detected by immunodetection (Fig 4D and 4E). The CV and F1 activities were also increased in the complemented mutant (Fig 4G), indicating that over-expression of C1-FDX-GFP leads to accumulation of CV, F1, and α/β. In contrast, the abundance of the F' module was decreased in the complemented mutant (Fig 4D and 4E). Notably, the C1-FDX-GFP fusion protein was hybridized at F' by immunodetection, implying that C1-FDX is associated with the F' module. To further test the presence of C1-FDX in F', the target BN gel band of F' was cut off from the wild-type, *c1-fdx-1*, and *c1-fdx-1*::C1-FDX-GFP, and analyzed by LC-MS/MS. C1-FDX was identified in high abundance in the F' band slices of the wild-type and *c1-fdx-1*::C1-FDX-GFP but not in the *c1-fdx-1* mutant (S1 Table). The Arabidopsis CV consists of 16 subunits [20]. Except for the c subunit of Fo, all the components were identified in this LC-MS/MS analysis (S1 Table). These results confirm that C1-FDX is associated with the F' complex, and the truncated C1-FDX loses this capability.

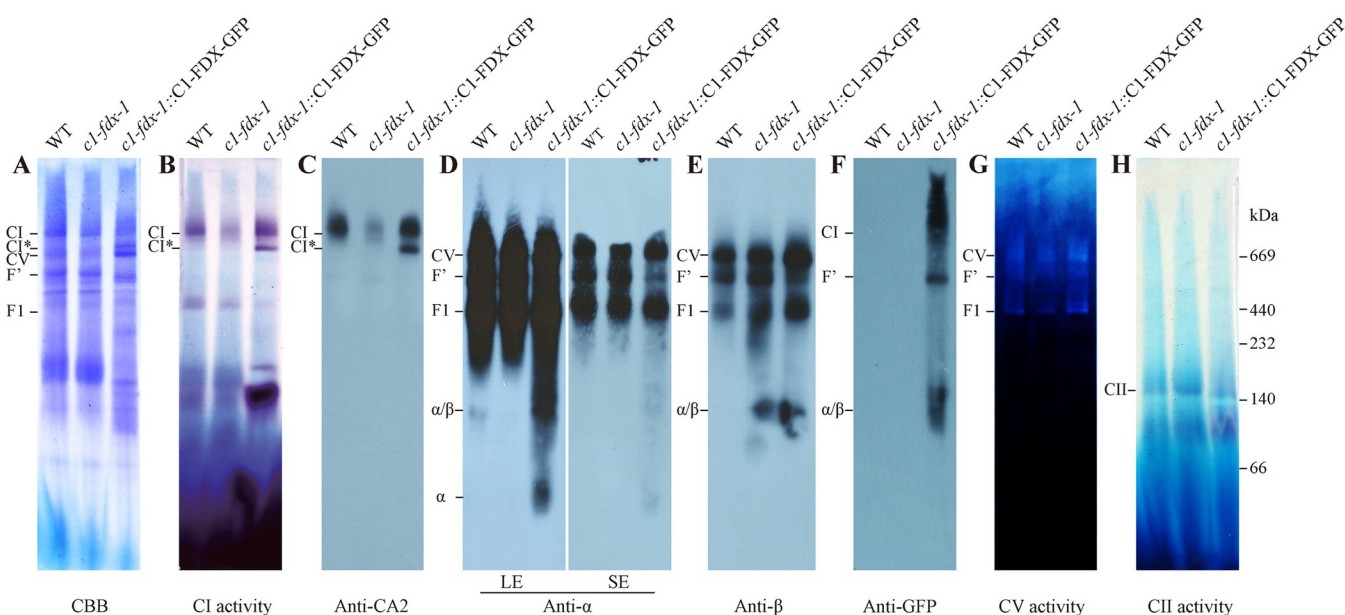

**Fig 4. Over-expression of C1-FDX increased the assembly of CI and CV.** Mitochondria were solubilized with 2% dodecylmaltoside. Mitochondrial membrane complexes were separated by 4–16% BN-PAGE, followed by CBB staining (**A**), in-gel activity assays of CI (**B**) and CV (**G**), and immunoblotting assays (**C-F**). The antibodies against CA2 and α, β subunits were used to detect the abundance of CI and CV, respectively. SE, short-time exposure; LE, long-time exposure. The GFP monoclonal antibody was used to detect the abundance of C1-FDX-GFP. CII activity was used as the loading control (**H**).

In addition, C1-FDX-GFP was also detected in the α/β sub-complex (Fig 4D and 4E). Compared with the wild-type that showed a weak α subunit signal at the α/β sub-complex (Fig 4D), the α subunit signal in the α/β sub-complex was decreased in *c1-fdx-1*, while increased in the *c1-fdx-1*::C1-FDX-GFP transgenic line as the long exposure detected a strong α subunit signal. Hybridization with anti-β antibody showed a low abundance of the β subunit in the wild-type and increased abundance in the *c1-fdx-*1 mutant and the C1-FDX-GFP transgenic line (Fig 4E). The results suggest that the α:β stoichiometric ratio in the *c1-fdx-1* is altered compared with that in the wild-type. The mutation of *C1-FDX* appears to lead to an increased ratio of β subunit in the α/β sub-complex. The activity of CII was not changed between *c1-fdx-1*::C1-FDX-GFP transgenic lines and the wild-type (Fig 4H).

## C1-FDX interacts with multiple subunits of CI and CV

The cryo-EM structure of CI shows that C1-FDX forms a ferredoxin bridge (A6:ACP2:C1-FDX) with ACP2 linking the matrix and membrane arm in Arabidopsis [2] and *Tetrahymena* mitochondria [12]. To identify the proteins that may interact with C1-FDX during the assembly process, we first investigated the interaction of C1-FDX and ACP2 with relevant CI subunits using the yeast two-hybrid (Y2H) assay. The results showed that C1-FDX interacted with ACP2, A6, B9, and four CA module subunits (CA1, CA2, CA3, and CAL1) in the Y2H assay (Fig 5A), with strong interaction with the Q module subunit A6 and the $P_D$ module subunit B9. ACP2 interacted with A6 in Y2H. This result conforms with A6:ACP2:C1-FDX forming a ferredoxin bridge. In Arabidopsis, ACP1 is also a subunit of CI, which is attached to the B9 subunit [2]. Arabidopsis ACP1 and ACP2 are acyl carrier proteins sharing a 53.97% identity in amino acid sequences (S9 Fig). Thus, we tested ACP1 as well. The result showed that C1-FDX also interacted with ACP1 in yeast but weakly (Fig 5A). ACP1 and ACP2 showed interaction only with CAL1. Considering the interchangeability of CAL1 and CAL2 in the CA domain and a 90% sequence identity between the two proteins [32], the different behavior of CAL1 and CAL2 may be due to different properties of the two proteins in yeasts. CA1, CA2, and CAL2 contact each other in the CI complex [2]. However, CA1 and CA2 did not show interaction in the yeast two-hybrid assay [33].

Similarly, we analyzed the interaction of C1-FDX with the F1 subunits (α, β, γ, δ, and ε) and the three Fo subunits of CV using Y2H. The results showed that C1-FDX could interact with four F1 subunits β, γ, δ, and ε, but not with the α subunit of F1 and the three Fo subunits (Fig 5B). ACP1 interacted with the δ subunit, while ACP2 did not interact clearly with the F1 components. To test whether the truncated c1-fdx-1 protein still poccesses the function to interact with the proteins that full-length C1-FDX interacts with, we tested the truncated c1-fdx-1 using the Y2H assay. The result showed that the truncated c1-fdx-1 protein could not interact with any of these subunits (S10 Fig), indicating that the truncated c1-fdx loses its protein-protein interaction function.

To verify the interactions detected by the Y2H, we employed the bimolecular fluorescence complementation (BiFC) and co-immunoprecipitation assays (Co-IP). Strong YFP signals were detected in mitochondria with combinations of C1-FDX-cYFP with the candidate proteins (Fig 5C), and no YFP signals were found in the negative control (S11 Fig), indicating that C1-FDX interacted *in vivo* with the CI component B9, the F1 components β, γ, δ, and ε. We further tested these interactions using Co-IP. Consistent with the results of Y2H and BiFC, the interacting proteins B9, β, γ, and ε co-immunoprecipitated with C1-FDX (Fig 5D). Co-IP analysis was not performed on the δ subunit because the recombinant δ protein could not be expressed. Instead, we used the LCI assay, and the result confirmed that C1-FDX interacted with the δ subunit (S12 Fig). Thus, the interactions between C1-FDX and B9 and the β, γ, δ,

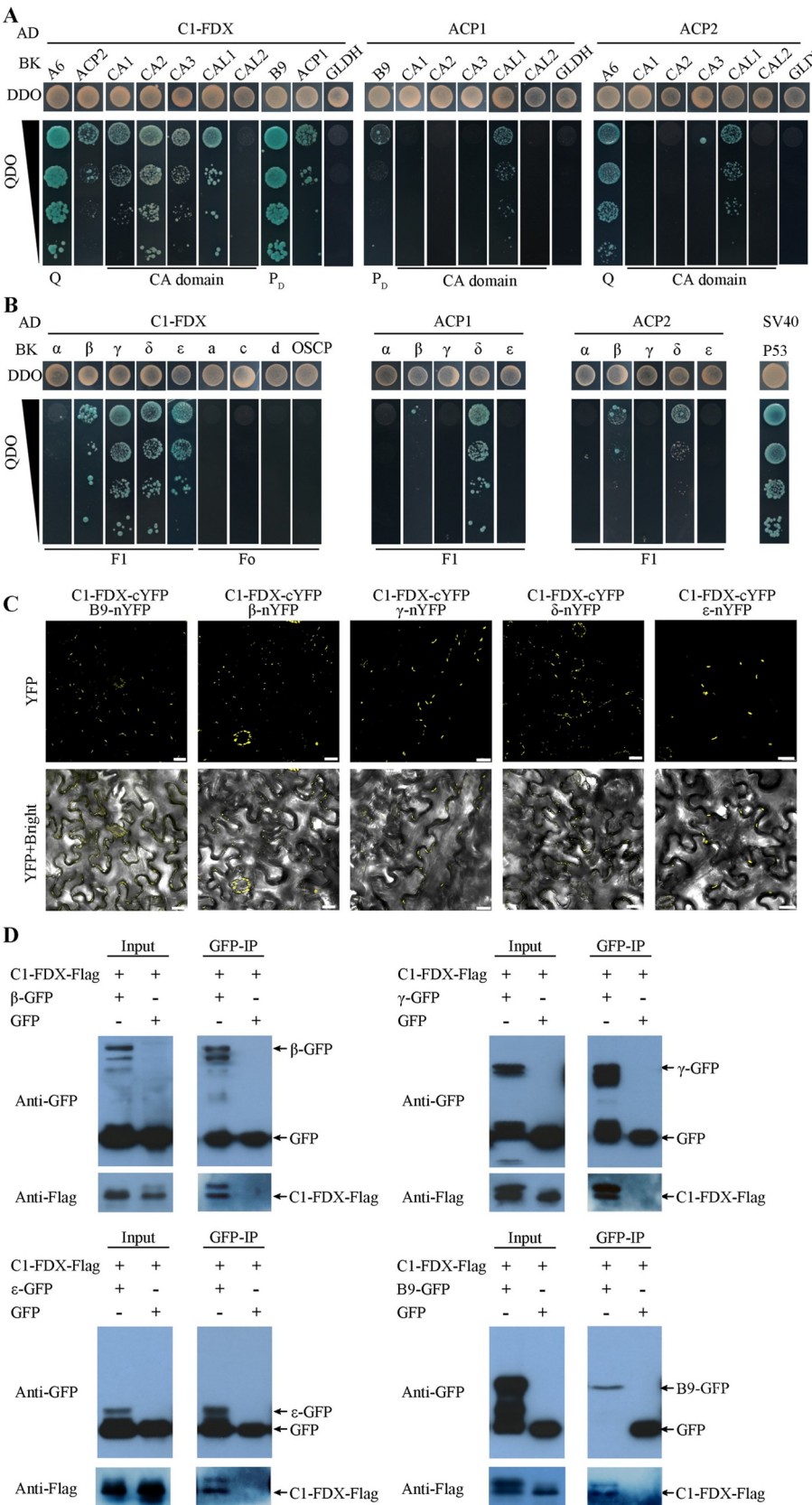

**Fig 5. C1-FDX and ACP1 or ACP2 interact with the multi-subunits of CI and CV.** Yeast two-hybrid interaction assay was carried out using C1-FDX, ACP1, and ACP2 cloned into the tagged pGADT7 vector. CI **(A)** and CV subunits **(B)** were cloned into the pGBKT7 vector. All the recombined vectors were transformed into the AH109 yeast strain. All positive interactions were grown on double drop-out (DDO) medium (SD-Leu-Trp) and serially diluted quadruple drop-out (QDO) medium (SD-Leu-Trp-Ade-His). The interaction of pGAD-SV40 and pGBK-p53 was used as a positive control. **(C)** C1-FDX interacts with B9, β, γ, δ, and ε using the bimolecular fluorescence complementation (BiFC). BiFC assay in tobacco leaves shows an interaction between C1-FDX and B9, β, γ, δ, ε subunits. Bar, 20 μm. **(D)** C1-FDX interacts with B9, β, γ, and ε using co-immunoprecipitation (Co-IP). The GFP tag was used as the negative control. The arrows denote the target fusion proteins.

and ε subunits of F1 were tested by three independent approaches. These results indicate that C1-FDX plays a role in the assembly of CV.

## Discussion

### C1-FDX is involved in the assembly of CI* in Arabidopsis

Recent studies have shown that C1-FDX forms a ferredoxin bridge (A6:ACP2:C1-FDX) with A6 and ACP2 connecting the membrane arm with the matrix arm of CI [2,10,11], and is responsible for the formation of supercomplex CI+III$_2$ [13]. The iron ion of C1-FDX may assist the proton migration toward the ND2 half-channel entrance by a positive repulsive electrostatic force [34]. In this study, we found that C1-FDX is also involved in the assembly of CI*. Over-expression of C1-FDX-GFP increased the abundance of CI* while no other assembly precursors of CI* were accumulated (Fig 4C). The activity of CI* is also increased in the *c1-fdx-1*::C1-FDX-GFP complemented mutant (Fig 4B). This result implies that C1-FDX, as a component of the ferredoxin bridge protein, could promote formation of the ferredoxin bridge and then increase the assembly of CI*. In contrast, the loss of C1-FDX decreased the abundance of CI*, while its assembly precursor P$_P$ module was accumulated (Fig 3A), implying the assembly process of P$_P$ to CI* is blocked. In addition, the N module was also accumulated in the *c1-fdx-1* mutant (Fig 3A). The N module consists of the CI matrix arm, which assembles with the P$_P$ module to form CI* in the CI assembly process [15]. These results confirm that C1-FDX is required for the CI* assembly. In addition, the loss of C1-FDX inhibits the assembly of CI as CI* assembles with the P$_D$ module to form CI in Arabidopsis [15]. In this study, we found that the activity and abundance of CI and CI* decreased in the *c1-fdx-1* mutant. In contrast, the P$_D$ module accumulated (Fig 3A), suggesting that the decreased abundance of CI might result from the decreased CI* formation (Fig 3A).

We found evidence for the potential formation of a ferredoxin bridge linking the P$_D$ and P$_P$ modules in the Arabidopsis CI. This point is based on the strong interaction between C1-FDX and B9 in the P$_D$ domain in the Y2H, BiFC, and Co-IP assays (Figs 5A, 5C, 5D and 6). Meanwhile, ACP1, a protein anchored to B9, also interacts with CAL1 (Figs 5A and 6). These results suggest that C1-FDX links the P$_P$ and P$_D$ modules via forming a B9:ACP1:C1-FDX:CAs bridge (ferredoxin bridge domain) (Fig 6). In addition, P$_P$ and P$_D$ modules, as well as CI assembly factor GLDH (preventing P$_D$ to P$_P$), increase in abundance in the *c1-fdx-1* mutant, while CI* is the opposite (Fig 3A). Based on these results, the block in assembling P$_D$ to CI* to produce CI might be caused by the lack of the second ferredoxin bridge. A recent study shows that C1-FDX and the CA domain are important for the efficient assembly of the P$_D$ module in the CI assembly process in Arabidopsis [13]. This slice of evidence supports our point that C1-FDX might form a ferredoxin bridge domain to affect the assembly of the P$_D$ module. However, the second ferredoxin bridge or C1-FDX at the B9 position is not detected in the cryo-EM structures of CI and supercomplex CI+III$_2$ [2,10,11]. It is possible that our detected interactions are an artifact, or these interactions are weak, and C1-FDX fell off during the

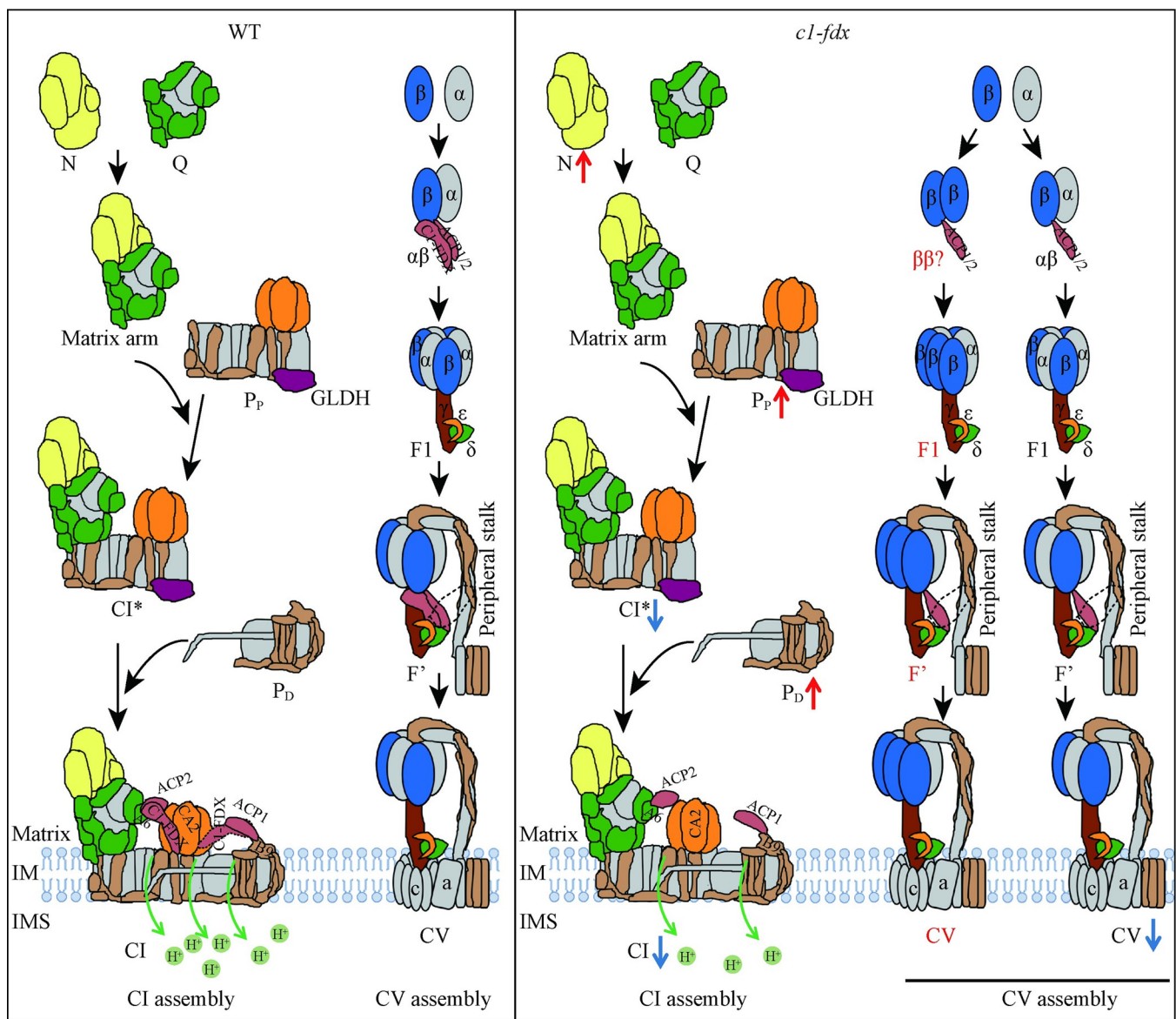

**Fig 6. A proposed model for the role of C1-FDX in the assembly of CI and CV.** N, N module; Q, Q module; $P_P$, $P_P$ module; $P_D$, $P_D$ module; IM, inner membrane; IMS, intermembrane space. The upward red arrows indicate an increase in abundance. The downward blue arrows indicate a decrease in abundance. The black dotted frame indicates the putative assembly factors. The C1-FDX marked with a black dotted frame between the B9 and CA2 subunits indicates the putative second bridge. The red letters indicate the putative incorrect assembly intermediates. The assembly intermediates and pathways of CI are based on the description by Ligas et al. [15].

solubilization process. Another possibility is that this bridge might be formed under a certain condition.

## C1-FDX may promote the subcomplexes assembly of CV in Arabidopsis

Although several factors with functions in different steps of the CV assembly have been identified in bacteria, yeast, or mammals [35], few have been identified and characterized in plants. In this study, we show that C1-FDX not only serves as a structural component of CI but also possesses the function of an assembly factor in the subcomplexes assembly of CV during seed

germination in Arabidopsis mitochondria. Three pieces of evidence support this conclusion. First, a loss of the C1-FDX function decreases the CV activity in 3-days-seedlings, especially the activity of the F1 sub-complex (Figs 2A and 6), and reduces the production of ATP in mitochondria (Figs 2C and 6). This impact on the CV complex and ATP level is observed in CV assembly factors in humans and yeast, for example, ATP11, ATP12, and ATP23 [24,36,37]. On the other hand, over-expression of C1-FDX promotes the assembly of CV and the sub-complexes F1 and α/β (Fig 4D and 4E). Second, we show that C1-FDX is physically associated with the CV assembly intermediates sub-complexes F' and α/β during CV assembly (Fig 4E and S1 Table). C1-FDX is disassociated from the mature CV once formed, as C1-FDX is not detected in the mature CV. Third, C1-FDX showed interaction with the β, γ, δ, and ε subunits of the F1 sub-complex *in vitro* and *in vivo* (Figs 5B and 5D, S12, and 6). Taken together, the physical interaction with the CV subunits, the association with the F' and F1 sub-complexes, and the genetic data demonstrate that C1-FDX is involved in the subcomplexes V assembly process. In Arabidopsis, AtATP11 and AtATP12 were recently identified as the CV assembly factors, which could interact with the β and α subunits, respectively [38].

Furthermore, this study presents evidence that C1-FDX may be involved in the assembly of the α/β sub-complex in Arabidopsis. C1-FDX was found to be associated with the α/β sub-complex (Fig 4F). In addition, over-expression of C1-FDX-GFP promotes the accumulation of α/β (Fig 4D and 4E). In contrast, the loss of the C1-FDX function increases the β subunit abundance in F1 and F', altering the strict 1:1 stoichiometric ratio in the $\alpha_3\beta_3$ hexamer, while the Fo subunit a is not changed (Figs 3B and 6). Assembly of the $\alpha_3\beta_3$ hexamer involves assembly factors. In yeast, chaperones ATP11 and ATP12 are identified as the CV assembly factors as ATP11 and ATP12 bind to the interface of the α and β subunit to prevent the formation of the insoluble α-α and β-β homodimers [23,39]. Subunits α and β possess the property to form homodimers without the assembly factors. Whether the assembly of the $\alpha_3\beta_3$ hexamer needs other factors in plants is unknown. In the *c1-fdx* mutant, the loss of C1-FDX may result in an incorrect assembly of the α and β subunits in the $\alpha_3\beta_3$ hexamer with excess β subunits and fewer α subunits (Fig 4D and 4E). C1-FDX interacted with the β subunit (Fig 5B–5D), implying that it could affect the binding between α and β. In yeast, the γ subunit initiates the release of Atp11 and Atp12 from the β and α subunits because the heterodimer of Atp11/β and Atp12/α can be isolated from the *Δγ* mutant, but the α/β heterodimer cannot [26]. A recent study of *in vitro* CV assembly in *Acetobacterium woodii* showed that a single subunit α or β could bind to γ or γε, forming αγ, αγε, βγ, βγε, respectively [40], indicating that the γ subunit plays a key role in the assembly of α or β subunits. C1-FDX showed strong interaction with the subunit γ, δ, and ε in Arabidopsis (Fig 5B–5D). Thus, the loss of C1-FDX might block the function of the γ subunit to release ATP11 or other potential assembly factors from the β subunit in Arabidopsis.

C1-FDX plays a role in the assembly of the F' sub-complex (Fig 4F and S1 Table). The F' sub-complex is a sub-complex of CV identified in maize [27,28]. The assembly and the exact composition of F' is unknown in plants. Over-expression of C1-FDX-GFP decreases the F' abundance, implying that the C1-FDX promotes F' to assemble into CV (Fig 4D and 4E). In humans, two assembly pathways of the Fo module are found [22]. In the first assembly pathway, the F1 module assembles to c-ring to form the S1 sub-complex, then combines with the peripheral arm (such as subunits OSCP, b, d, F6) to form CV, which is similar to that in Arabidopsis [20]. In the second pathway, the F1 module first combines with the peripheral arm and then integrates into the c-ring to form the CV. This assembly pathway is similar to that in yeast [21]. In our study, 15 out of 16 CV subunits were detected at the F' position by LC-MS/MS in the wild-type, *c1-fdx-1*, and *c1-fdx-1*::C1-FDX-GFP, except for the c subunit (S1 Table). There is a possibility that the c-ring is not a component of F', which is similar to the

intermediate of the second assembly pathway of CV composed of the F1 module and peripheral arm [22]. In this case, the F' sub-complex may combine with the c-ring to form CV in Arabidopsis. However, we did not find the interaction between C1-FDX and the c subunit and the peripheral arm subunits, such as a, d, and OSCP (Fig 5B). Instead, we found that C1-FDX interacts with the F1 subunits β, γ, ε, δ, and ACP1, where ACP1 strongly interacts with the δ subunit, implying that C1-FDX and ACP1 are possibly involved in the recruitment of c-ring with other assembly factors. In addition, we speculate that the involvement of C1-FDX in the assembly of α/β and F' are two separate processes (Fig 6) because C1-FDX is not present in the F1 module. In yeast, ATP23 is also involved in two processes during the CV assembly; one is the combination of subunits a and 8, and the other is the assembly of the peripheral arm with the c-ring/F1 [37].

## C1-FDX plays a role in seed germination of Arabidopsis

It has been reported that a loss of C1-FDX did not cause any discernible morphological alterations in the Arabidopsis plants with a compromised CI [13,41]. Indeed, the *c1-fdx* mutants in this study displayed no visible difference from the wild-type under standard conditions (S3A and S3B Fig). However, the seed germination was delayed by 24 h in the *c1-fdx* mutant compared with the wild-type (Fig 1C). C1-FDX is rapidly expressed 24 h after stratification (S3D Fig). The genes of respiration chain complex subunits are rapidly expressed after stratification and reach the maximum at 24 h in Arabidopsis [42]. This process is consistent with mitochondrial maturation during seed germination in rice [43] and maize [44]. The mitochondria in mature, dry seeds are dysfunctional in structure and composition, lacking the cristae and respiratory chain complexes, thus termed promitochondria [45]. After stratification, the respiration chain complexes are rapidly synthesized and assembled, and cristae are formed, converting promitochondria to mature mitochondria [42,46]. The CI and CV assembly requires the function of C1-FDX; thus, the absence of C1-FDX impacts this process in the early germination of Arabidopsis. On the contrary, overexpression of C1-FDX-GFP leads to increased amounts of complex I and activity in this stage. As an indicator of the impact on CI and CV assembly, the ATP levels in the *c1-fdx* mutant were reduced by ~40% compared to the wild-type (Fig 2C). This may also account for the slow root growth in the *c1-fdx* seedlings (Fig 1C). Intriguingly, the loss of C1-FDX does not have a discernible impact on the growth of Arabidopsis under normal growth conditions, given that C1-FDX is a structural component of CI [2] and plays a role in the assembly of CI and CV in Arabidopsis (this study). Whether there is a redundant function or the impact appears only under specific conditions awaits further study.

## Materials and methods

### Plant materials and growth

Arabidopsis accession Columbia-0 (Col-0) was used as the wild-type (WT). Arabidopsis seeds were surface-sterilized with 0.63% (v/v) NaClO containing 0.2% (v/v) Triton-X 100 for 15 min and washed 4 times in sterile water. The sterilized seeds were spread on plates containing the Murashige and Skoog (MS) medium, 1% (w/v) sucrose, and 0.8% (w/v) agar or 0.4% (w/v) gelzan. The plates were kept in the dark at 4°C for 3 days before being transferred to growth chambers at 22°C with 70% humidity and a 16 h light/8 h dark cycle with light intensity of 80 μmol quanta m$^{-2}$ s$^{-1}$. After one week, the seedlings were transplanted to the soil and grew in a greenhouse under a long-day photoperiod of 16 h of light (100 μmol quanta m$^{-2}$ s$^{-1}$, 22°C, 68%-73% humidity).

For carbon deficiency tests, 1x MS medium containing 0.8% (w/v) agar without sucrose was used. For nitrogen and iron deficiency treatment, 1x MS medium without nitrogen or iron (Solarbio Life Sciences, China) was used but supplemented with 1% (w/v) sucrose and 0.8% (w/v) agar. For salt or mannitol stress treatment, 1x MS medium contained 1% (w/v) sucrose and 0.8% (w/v) agar and supplemented with 100 mM NaCl or 250 mM mannitol. The plates were kept in the dark at 4°C for 3 days before being transferred to growth chambers at 22°C with 70% humidity and a 16 h light/8 h dark cycle with the light intensity of 80 μmol quanta $m^{-2}$ $s^{-1}$ or dark condition. The seedling phenotypes were observed and photographed at 3 days after stratification.

## CRISPR/Cas9 mutagenesis of *C1-FDX*

The candidate CRISPR/Cas9 targets in *C1-FDX* were analyzed using CRISPR-P 2.0 (http://crispr.hzau.edu.cn/cgi-bin/CRISPR2/SCORE). Two pairs of target primers with high scores and probably knocking out the function of C1-FDX were chosen (S2 Table). The PCR fragments were amplified from pCBC-DT1T2 using the two pairs of primers and cloned into vector pHEC401. The pHEC401-2gR vector was transformed into *Agrobacterium* strain GV3101 and infected the inflorescence of wild-type Arabidopsis according to the floral dip method [47]. Seeds from the T0 plants were screened on MS plates containing 50 mg/L hygromycin. The resistant seedlings (T1) were transferred to the soil. The homozygous line without Cas9 was isolated and used for further analyses.

## Genetic complementation

*C1-FDX* was placed under the control of the 35S CaMV promoter in pGWB5 and transformed the *c1-fdx-1* and *c1-fdx-2* mutants. The full-length ORF of *C1-FDX* was amplified and then cloned into p-Super 1300 without the GFP sequence to generate p-Super 1300:*C1-FDX*. This construct and pGWB5:*C1-FDX* were introduced into *Agrobacterium tumefaciens* EHA105 and transformed the *c1-fdx-1* and *c1-fdx-2* mutants via the floral dip method [47]. The transgenic lines were screened on the MS medium containing 50 mg/L hygromycin.

## RNA extraction and quantitative real-time (qRT)-PCR

Total RNA was extracted from 100 mg tissue using the RNAPrep Pure Plant Kit (Tiangen, China) according to the manufacturer's instructions. RNA was transcribed into cDNAs with Hifair III 1st Strand cDNA Synthesis SuperMix (Yeasen, China). For RT-PCR, reactions were performed using the 2× Taq Master Mix (Vazyme, China). For qRT-PCR, reactions were performed using AceQ Universal SYBR qPCR Master Mix (Vazyme, China). The Arabidopsis actin gene *AtActin1* was used as the internal control. The relative gene expression value was calculated with the $2^{-\Delta\Delta Ct}$ method [48]. For each sample, three biological replicates were analyzed. The RT- and qRT-PCR primers are listed in S2 Table.

## Mitochondrion isolation and protein complexes extraction

After surface sterilization, 500 μg seeds of each sample were cultured in 150 mL liquid 1/2 MS medium containing 0.5% (w/v) sucrose in a flask mounted in a shaker with 45 rpm in the dark for 3 days at 25°C. At this stage, the growth of *c1-fdx* etiolated seedlings was slower than the wild-type. Crude mitochondria were isolated from the etiolated seedlings as described [49]. The mitochondrial membrane protein complexes were solubilized with 2% dodecylmaltoside (DDM) (Sigma, USA) or 4% digitonin (Sigma, USA) as described [17,50]. After solubilization, the samples were centrifuged at 22,000 g for 30 min. The supernatant was the membrane

protein complexes extract. The protein concentration was determined by the Bradford method [51].

## Blue native PAGE and in-gel activity assays

130 μg of membrane protein complexes extract were mixed with 6.8 μl sample buffer and 0.5 μl G-250 buffer of the Native PAGE Sample Prep kit (Thermo Fisher Scientific, USA). The mixed samples were loaded onto a 3.5–12% (w/v) or 4–16% (w/v) blue native PAGE (BN-PAGE) (Thermo Fisher Scientific, USA), and the electrophoresis was carried out according to the manufacturer's instruction. After electrophoresis, the blue native gels were stained by Coomassie brilliant blue. In-gel activities of CI, CII, CIII, CIV, and CV were assayed as described [52]. Image J 1.36 (http://rsb.info.nih.gov/ij/) was used to quantify the in-gel activity.

## Measurement of mitochondrial ATP levels

As mentioned, 50 μL of lysis buffer from the ATP assay kit (Beyotime, China) was added to 500 μL isolated mitochondrial suspension. The mixture was vortexed for 1 min and centrifuged at 12,000 g for 5 min at 4°C. After measuring protein concentration, the supernatant was used for the ATP test. 20 μL supernatant and 100 μL ATP detection buffer were mixed and then transferred to a 96-well microplate for luminescence measurement using a microplate spectrophotometer (Tecan, Switzerland). Each sample was analyzed with four biological replicates.

## Measurement of CV activity

The CV activity was measured using the Mitochondrial Complex V Activity Assay Kit (Solarbio Life Sciences, China) according to the manufacturer's instructions. Briefly, 3-day-old seedlings were treated with 10 μM oligomycin A and $H_2O$ (control) for 3 h. Then, 10 g of fresh seedlings after treatment were used for isolation of the mitochondria as described [49]. The mitochondria were sonicated using an ultrasonic cell crusher, and the protein concentration was determined using the Bradford method [51]. Each sample equivalent to 5 μg protein was used per reaction. Absorbance at 660 nm was used to calculate the CV activity. Each sample contains four biological replicates.

## Immunoblotting analysis

For immunodetection of mitochondrial membrane complexes, BN gel strips were cut out and soaked in a denaturing solution (1% SDS, 1% mercaptoethanol) for 1 h. Then, the proteins were transferred to the polyvinylidene fluoride (PVDF) membrane under 120V for 1 h at 4°C. The mitochondrial complexes were analyzed by immunoblotting with specific antibodies. The antibodies against the maize CI subunits Nad9, V1, A5, B10, and CV subunit α were produced from ABclonal Technology (China). The monoclonal antibody GFP was purchased from Thermo Fisher Scientific (USA). The rest of the antibodies against Arabidopsis mitochondrial proteins were performed as described [30]. All antibodies used were diluted at 1:1000. The secondary antibody was anti-rabbit IgG (1:10000) (ABclonal Technology, China) and anti-mouse IgG (1:5000) (ABclonal Technology, China). Immunodetection was carried out with the Western Bright ECL Kit (Advansta, USA).

For immunoblotting analysis of mitochondrial components, 10 μg of total mitochondrial proteins were separated by 12.5% SDS-PAGE. The proteins were transferred to the PVDF membrane and detected with specific antibodies. The antibodies against the maize CI subunits

Nad9, V1, A5, B10, and CV subunit α were produced from ABclonal Technology (China). The rest of the antibodies against Arabidopsis mitochondrial proteins were performed as described [30]. All antibodies used were diluted at 1:1000. The secondary antibody was anti-rabbit IgG (1:10000) (ABclonal Technology, China) and anti-mouse IgG (1:5000) (ABclonal Technology, China). Immunodetection was carried out with the Western Bright ECL Kit (Advansta, USA). The band intensity was measured using the software Image J 1.36.

## Protein identification by LC-MS/MS

The BN gel bands of F' in the wild-type, *c1-fdx-1*, and *c1-fdx-1*:: C1-FDX-GFP were prepared for mass spectrometry by in-gel tryptic digestion as described [50]. Mass spectrometric analysis was performed by Allwegene Technology (China). The enzymatic hydrolysate was eluted on a C18 Cartridge column (Sigma, USA). Peptides eluted from the column were sprayed using the Q-Exactive Plus mass spectrometer (Thermo Fisher Scientific, USA). Mass spectrometry raw data were analyzed using MaxQuant (v1.6.14).

## Yeast two-hybrid (Y2H) assay

The yeast two-hybrid assay was conducted according to the Matchmaker Gold Yeast Two-Hybrid System (Clontech, Japan). Full-length coding fragments of genes were cloned into the GAL4 activation plasmid (pGADT7) and GAL4 binding plasmid (pGBKT7) using restriction enzyme-mediated ligation, respectively. Protein-protein interactions were determined by growing the co-transformed yeast strains on SD/-Trp/-Leu/-His/-Ade/+ x-α-gal (QDO+ x-α-gal) medium for 4 days at 30˚C.

## Bimolecular fluorescence complementation (BiFC) assay

The full-length coding fragments of *C1-FDX*, *B9*, *β*, *γ*, *δ*, and *ε* were cloned into the split-YFP destination vectors (N- and C- termini), respectively [53]. The constructs were transformed into *Agrobacterium* strain EHA105 (Tsingke, China) and co-infiltrated the tobacco leaves. YFP fluorescence was observed 48 h after infiltration using the Carl Zeiss LSM880 microscope (Zeiss, Germany) with excitation at 500 nm.

## Co-immunoprecipitation (Co-IP) assay

The Co-IP assay was performed as previously described with minor modifications [54]. The full-length cDNAs of *C1-FDX* and *B9*, *β*, *γ*, *ε* were cloned into the p-Super 1300-Flag and p-Super 1300-GFP vectors, respectively. The resulting plasmid transformed into *Agrobacterium tumefaciens* EHA105 (Tsingke, China). The tobacco leaves were harvested 48 h after co-infiltration and ground in liquid nitrogen. Total proteins were extracted in 1 ml extraction buffer. The lysates were centrifuged at 12,000 g for 20 min. The supernatant was incubated with 20 μL of GFP-Nanoab-Magnetic Beads (LABLEAD, China). After 2 h of incubation at 4˚C, the beads were centrifuged and washed six times with extraction buffer. Proteins were eluted with 30 μL of 2× sample buffer and analyzed by immunoblotting using anti-GFP (ABclonal Technology, China) and anti-Flag antibodies (Thermo Fisher Scientific, USA).

## Luciferase complementation imaging assay (LCI)

The LCI assay was performed as described [55]. The cDNAs of C1-FDX and δ were cloned into pCAMBIA1300-nLUC and pCAMBIA1300-cLUC, respectively. The respective constructs were transformed into Agrobacterium strain EHA105 (Tsingke, China) and then co-infiltrated the tobacco leaves. After 48 h infiltration, the leaves were immersed in 1 mM D-luciferin

(Biosynth, Switzerland). The LUC activity was captured using the Lumazone FA Pylon 2048B system.

## Statistical analysis

All data are the means ± SD from at least three replicates. Statistical analysis was performed using GraphPad Prism 6.01 (USA). One-way ANOVA of the *t*-test was used to analyze the data. Differences with $P < 0.05$ and $P < 0.01$ were indicated as statistically significant (*) and extremely significant (**), respectively.

## Supporting information

**S1 Fig. The transcript sequence of *C1-FDX* in *c1-fdx-1* and *c1-fdx-2* mutant lines.**
(TIF)

**S2 Fig. The phenotype of *c1-fdx* at 3 days after stratification under stress conditions in Arabidopsis.**
(TIF)

**S3 Fig. The phenotype of *c1-fdx* and expression levels of *C1-FDX* during germination in Arabidopsis.**
(TIF)

**S4 Fig. The activity of CI and CV was reduced in the *c1-fdx-2* mutant.**
(TIF)

**S5 Fig. Effect of oligomycin A on CV activity in the *c1-fdx* mutant.**
(TIF)

**S6 Fig. Characterization of the specificity of the antibodies against CI and CV subunits in Arabidopsis.**
(TIF)

**S7 Fig. Loss of C1-FDX affects the mitochondrial protein levels.**
(TIF)

**S8 Fig. The expression of *AOX* genes in the wild-type and *c1-fdx* mutants by qRT-PCR.**
(TIF)

**S9 Fig. Sequence alignment of Arabidopsis ACP1 and ACP2.**
(TIF)

**S10 Fig. The truncated c1-fdx-1 protein could not interact with the C1-FDX interacted proteins.**
(TIF)

**S11 Fig. Negative control of the interaction between C1-FDX and B9, β, γ, δ, ε using the BiFC.**
(TIF)

**S12 Fig. C1-FDX interacts with the δ subunit using Luciferase complementation imaging (LCI) assay.**
(TIF)

**S1 Table. Identification of the F' composition by LC-MS/MS.**
(XLSX)

**S2 Table. Primers were used in this study.**
(DOCX)

## Acknowledgments

We thank Haiyan Yu, Xiaomin Zhao, Yuyu Guo, and Sen Wang from the Core Facilities for Life and Environmental Sciences of SKLMT (State Key Laboratory of Microbial Technology, Shandong University) for assistance in microscopy imaging of laser scanning confocal microscopy analysis. We also thank Professor Qi-Jun Chen from China Agricultural University for providing the vectors pCBC-DT1T2 and pHEC401.

## Author Contributions

**Conceptualization:** Baoyin Chen, Bao-Cai Tan.

**Data curation:** Baoyin Chen, Junjun Wang.

**Formal analysis:** Baoyin Chen, Junjun Wang, Manna Huang, Yuanye Gui, Qingqing Wei, Le Wang.

**Funding acquisition:** Baoyin Chen, Bao-Cai Tan.

**Investigation:** Baoyin Chen, Junjun Wang, Bao-Cai Tan.

**Methodology:** Baoyin Chen, Junjun Wang, Manna Huang, Yuanye Gui, Qingqing Wei, Le Wang, Bao-Cai Tan.

**Project administration:** Junjun Wang, Bao-Cai Tan.

**Resources:** Baoyin Chen, Junjun Wang, Bao-Cai Tan.

**Supervision:** Bao-Cai Tan.

**Validation:** Baoyin Chen, Junjun Wang, Manna Huang, Yuanye Gui, Qingqing Wei, Le Wang, Bao-Cai Tan.

**Visualization:** Baoyin Chen, Junjun Wang, Bao-Cai Tan.

**Writing – original draft:** Baoyin Chen, Bao-Cai Tan.

**Writing – review & editing:** Baoyin Chen, Junjun Wang, Bao-Cai Tan.

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
