## [Decision Letter · Decision Letter 0]

9 Jan 2024

Dear Dr Tan,

Thank you very much for submitting your Research Article entitled 'C1-FDX is required for the assembly of mitochondrial complexes I and V in Arabidopsis' to PLOS Genetics.

The manuscript was fully evaluated at the editorial level and by independent peer reviewers. The reviewers appreciated the attention to an important problem, but raised some substantial concerns about the current manuscript. Based on the reviews, we will not be able to accept this version of the manuscript, but we would be willing to review a much-revised version. We cannot, of course, promise publication at that time.

If you decide to revise the manuscript for further consideration at PLOS Genetics, please aim to resubmit within the next 60 days, unless it will take extra time to address the concerns of the reviewers, in which case we would appreciate an expected resubmission date by email to plosgenetics@plos.org.

We are sorry that we cannot be more positive about your manuscript at this stage. Please do not hesitate to contact us if you have any concerns or questions.

Yours sincerely,

Prof. Dr. Oren Ostersetzer

Guest Editor

PLOS Genetics

Gregory P. Copenhaver

Editor-in-Chief

PLOS Genetics

Specific comments:

We have now completed the review of your manuscript “C1-FDX is required for the assembly of mitochondrial complexes I and V in Arabidopsis” (PGENETICS-D-23-01284). I apologize for the long editorial process, but initially we received two reviews with different recommendations and we had to consult a third referee. As you can see from the comments below, two of the referees (Ref’s 2 and 3) express considerable concerns over the work described in the manuscript. In particular, they feel that the experiments lack some important controls and that additional work is needed to support the main conclusions in the manuscript. All three referees make several good suggestions for improving the MS text, as well as the data, controls and additional experiments that should be included to provide further and more conclusive evidence for the analyses of the function of C1-FDX, a mitochondrial ferredoxin-like protein which is located in the so-called bridge module of mitochondrial complex I in plants, alga and Tetrahymena. Because the recommended work is more extensive and exceeds that what usually is requested for minor revisions, I have no alternative but to decline the acceptance of the manuscript at its current form/stage and request for major revisions. I would like to encourage you, however, to consider the three referee’s suggestions (in particular look at the comments of those of reviewers 2 and 3) and prepare a revised manuscript for PLOS Genetics that addresses the concerns and comments raised by the 3 referees.

Reviewer's Responses to Questions

**Comments to the Authors:**

Reviewer #1: The manuscript entitled “C1-FDX is required for the assembly of mitochondrial complexes I and V in Arabidopsis” presents an interesting investigation into the physiological function of the so-called Complex I ferredoxin (C1-FDX) by Baoyin Chen et al. The authors assert that C1-FDX serves as an essential assembly factor for mitochondrial complexes I and V. The manuscript is well-written, and the conclusions are supported by the experiments; however, I have some comments.

1-In line 65, the authors state, “In humans, mFDX1 and mFDX2 play an essential role in the assembly of the Fe-S cluster and the biosynthesis of heme A (Sheftel et al., 2010). However, the function of mFDX1 and mFDX2 in plants remains to be revealed.”

This statement may not be entirely accurate, as there are relevant papers that should be cited. For instance, the plant ADX–ADXR system has been linked to biotin synthesis in collaboration with BIO2, an iron–sulfur (Fe-S) cluster enzyme (Picciocchi et al., 2003). Additionally, Bellido AM et al. (2022) identified a mitochondrial ADXR–ADX–P450 electron transport chain crucial for maternal gametophytic control of embryogenesis in Arabidopsis. This chain, involving mFDXs (ADX1 and ADX2), is associated with steroid hormone synthesis.

2-The manuscript does not clearly specify the expected peptide or truncated C1-FDX. Including this information in Figure 1 or discussing the possibility of detecting this peptide via LC:MS/MS would be beneficial. It is also essential to consider potential interactions and functionality of the truncated peptide, which comprises a substantial portion of the wild-type protein.

3-Regarding BN gels, it is noted that the authors were unable to observe the supercomplex I + III2. C1-FDX may impact the stability of this supercomplex. I read in the M & M section that the authors use dodecylmaltoside or digitonin. In my experience, the supercomplex is only seen using digitonin but the gels that the authors show were solubilized with dodecylmaltoside. Have the authors seen any differences in complexes using digitonin?

4-The manuscript mentions that overexpressed C1-FXD-GFP leads to increased amounts of complex I and activity. An explanation for this phenomenon should be provided.

5-Concerning interaction experiments, the lack of interaction of CAL2 with any other protein raises questions. Clarification on the physiological significance of this observation is warranted, particularly considering the interchangeability of CAL1 and CAL2 in CI models.

6-The interaction between both ACP1 and ACP2 with CAL1 and not CAL2 requires an explanation of its physiological relevance.

7-BIFC experiments are deemed unconvincing, and troubleshooting with protocols is suggested to address the issue of observing only a few concentrated yellow dots in a particular zone.

8-While immunoprecipitations are clear, it is unclear why the interactions between C1-FDX and the CA proteins observed by Y2H were not confirmed.

9-Testing interactions with the truncated protein in addition to the full-length protein would add valuable insights and might explain some inconsistences.

10-The manuscript raises the question of why, in a defective C1-FDX background plants, exhibit still reduced amounts of complexes I and V. Exploration of redundant proteins that may compensate for C1-FDX deficiency (if any) is suggested.

Reviewer #2: Chen and co-workers present a study to elucidate the function of a mitochondrial ferredoxin, C1-FDX, which is located in the so-called bridge module of mitochondrial complex I in plants, alga and Tetrahymena. The bridge module links the carbonic anhydrase module with the ubiquinone-reduction domain of complex I. Using CRISPR-Cas9 methodology, two Arabidopsis C1-FDX mutants were generated. Both carry a point mutation causing a shortened open reading frame. Mutant plants show delayed germination and decreased levels of complex I and complex V, as shown by Blue-native (BN) PAGE in combination with in-gel activity measurements. In contrast, overexpression of C1-FDX leads to increased levels of complexes I and V. Finally, abundances of different subunits of the complexes I and V are investigated in wild-type and mutant strains, as well as protein-protein interactions using Y2H, bimolecular fluorescence complementation (BiFC) and co-immunoprecipitation assays (Co-IP). The authors conclude that C1-FDX directly interacts with complexes I and V and is important for the assembly of both complexes. The authors present a wide range of experiments. The presented results are interesting, but nevertheless I have a few questions and objections:

Figure 1: Is C1-FDX absent in the two CRISPR-Cas9-lines c1-fdx-1 and c1-fdx-2? Since the mutation is at position 338, truncated proteins of 113 amino acids (instead of 159 amino acids) could be translated and imported into the mitochondria (in the truncated proteins, the C-terminal 46 amino acids would be absent). The truncated genes are transcribed (Fig. S3C). Presence or absence of C1-FDX could be clarified by MS/MS analyses of complex I bands from a BN gel.

A corresponding experiment actually has been done and is presented in the end of the manuscript (Table S2): The F’-band of complex V from wild-type (wt)-plants and from the mutants c1-fdx-1 & c1-fdx-1::C1-FDX-GFP was cut out from a BN gel and analyzed by MS/MS. C1-FDX was found in wt plants but not in mutant c1-fdx-1 (please add the amino acid sequences of the identified peptides in table S2; please present a complete list of all identified proteins). However, in wt plants, only one single peptide of C1-FDX was detected (the protein coverage is only 5.7%). Unfortunately, presence of C1-FDX is therefore not well secured. C1-FDX is a hydrophilic protein and previously several peptides have been detected by MS/MS, e.g. 9 peptides in Klusch et al. 2023 (Supp. table 4; the resulting protein coverage is 60%). The low level of C1-FDX detection in wt plants could be interpreted as a C1-FDX background on the BN gel in the region of the F’-band of complex V.

In the future (not for the current manuscript), complexome profiling experiments of mitochondria from wt-plants and C1-FDX mutant plants should be carried out to systematically characterize C1-FDX-containing assembly intermediates of the complexes I and V.

Figure 2 and 3: Please indicate the protein amounts loaded per lane of the gels and that equal protein amounts were always loaded for wt and the mutants, respectively.

Figures 2-4: I can follow that absence or truncation of C1-FDX in c1-fdx-1 and c1-fdx-2 mutant lines affects abundance and activity of complex I and complex V. For complex I, this has been shown previously using two mutant C1-FDX Arabidopsis-lines obtained by the CRISPR-Cas9 methodology (Röhrich et al. 2023). In these mutants, which seem to be true deletions, complex V was not affected. The discrepancy of these results and the results obtained by the analysis of the c1-fdx-1 and c1-fdx-2 lines should be discussed.

Could complex V reduction in the c1-fdx-1 and c1-fdx-2 mutants be based on an indirect effect? Could it be that reduction in complex I in the c1-fdx-1 and c1-fdx-2 mutants results in a decreased proton gradient across the inner mitochondrial membrane, which causes downregulation of genes encoding subunits of the mitochondrial ATP synthase complex, which uses the proton gradient for ATP synthesis? Please comment. On the other hand, the authors show direct interaction of C1-FDX and other subunits of the complexes I as well as subunits of complex V by Y2H, BiFC and Co-IP (Figure 5). Some of the results are nicely in line with published structural data for Arabidopsis complex I, some others are surprising. I am not an expert on evaluating these results. Can false-positive results be excluded using these procedures? Please comment.

Minor points:

Introduction section, Line 48: “inter-outer membrane space”, rather write “intermembrane space”

Results section, lanes 156, 203, 214, 227, 269: please avoid “to verify…”, but rather use “to test…”, to demonstrate a neutral attitude with regard to your hypotheses.

Discussion section: There is a new hypothesis on C1-FDX function which you might want to consider for discussion (PubMed 37599162)

Reviewer #3: The manuscript “C1-FDX is required for the assembly of mitochondrial complexes I and V in Arabidopsis” describes the creation (using CRISPR-Cas9) and characterisation of Arabidopsis mutants impaired in the expression of C1- FDX, a component of a ferredoxin bridge known to be important for complex I assembly and supercomplex (I + III) formation.

The introduction is informative and well written, and leads to the question: what is the precise role of the C1-FDX protein in Arabidopsis? Experiments are well conducted, and the results mostly clearly shown, apart from the western blots on BN gels. This is a big issue because most of the conclusions on involvement of C1-FDX in assembly of Complex V are based on these experiments.

The main problem as far as I am concerned is in the discussion. Although the evidence in favour of C1-FDX acting as an assembly factor in complex I assembly is strong, I am less convinced about its role in Complex V assembly. The data from the BN PAGE gels and especially the western blot images are not easy to interpret as they often appear over-exposed, making the quantification of the bands rather unreliable.

The discussion concerning the pathways proposed is difficult to follow. It is not very clear probably because the evidence is not fully convincing and suffers from slight over-interpretation. Each stage of the assembly processes of both Complex I and Complex V should be described more clearly, referring to the model provided.

Major points:

- The signals on several western blot images in Figure 3A and Figure 4 D, E and F, including the short exposure images, seem saturated to me. If so, the quantification using Image J cannot be accurate.

- For example, line 178, the authors claim that CI* was also decreased in the mutant (Figure 3A). Although it is obvious with the anti-Nad9 antibody, it is a lot less clear with the anti-A5 antibody, and CI* is not detected with the anti-V1 antibody (Figure 3A), which I would expect to recognise CI* as it has the N module attached to it. Can the authors comment on that?

- In Figure 4A, it is obvious from the Coomassie blue staining (which is proportional to the quantity of protein in each band) that the mitochondrial protein pattern of the complemented line is very different from that of the WT and the mutant. Some major bands are missing from the complemented line, which in turn has many additional bands. This suggests that the “equal” mitochondrial protein loadings measured by Bradford assay are not equivalent in all lines, and that comparing quantities of the different sub-complexes between lanes is difficult because of that. That, in addition to the saturated signals and the poor resolution of BN gels in general, makes the interpretation rather unconvincing.

- Line 235, the authors state: ‘In addition, the C1-FDX-GFP was also detected in the α/β sub-complex (Figure 4D and E). Compared with the wild-type, the abundance of α subunit in the α/β sub-complex was decreased in c1-fdx-1, while the abundance of β subunit was increased, suggesting that C1-FDX may play a role in the α/β assembly. This may be the cause for the altered α:β stoichiometry in c1-fdx-1 (Figure 3B). Therefore, C1-FDX plays a role in CI and CV assembly during seed germination in Arabidopsis.’

I do not agree with that statement, because in Fig 4 E, the mutant and the complemented line both have similar signals for the so called α:β intermediate complex, which seems to be mostly comprising the β subunit. As for the complemented line, I would have expected a similar signal for the α and β subunits westerns, as it expresses high levels of C1-APX, but this is not the case. There is still a lot less α subunit in the so called α:β intermediate complex, suggesting that C1-APX is not needed for assembly of the α:β sub-complex.

Therefore, the authors should be less assertive in their conclusions concerning the role of C1-APX in the assembly of Complex V.

Minor points:

- In Figure 3, The MW marker sizes are not indicated.

- In Figure 4G, the Complex V activity bands are very difficult to see, so it is difficult to assess if CV, F’ or F1 are increased.

- The blue and black arrows in Figure 6 are not very easy to tell apart. I would suggest using different shapes for arrows showing the various steps in the assembly model (black arrows) and to indicate abundance variations (red and blue arrows).

Role of C1-APX in the assembly

In conclusion, I agree with the conclusions that C1-APX is involved in the assembly of Complex I because the evidence is strong, but I am more doubtful about what its involvement in Complex V is.

I think the results and discussion concerning this aspect needs to be rewritten if the authors present better western blots and want too fully convince the reader.

The manuscript should be improved in order to be convincing on this point or the authors should be less affirmative about the role of C1-FDX in Complex V assembly.

**Have all data underlying the figures and results presented in the manuscript been provided?**

Reviewer #1: Yes

Reviewer #2: Yes

Reviewer #3: Yes

PLOS authors have the option to publish the peer review history of their article (what does this mean?). If published, this will include your full peer review and any attached files.

Reviewer #1: **Yes: **Eduardo Zabaleta

Reviewer #2: **Yes: **Hans-Peter Braun

Reviewer #3: No

---

## [Decision Letter · Decision Letter 1]

29 Apr 2024

Dear Dr Tan,

Thank you very much for submitting your Research Article entitled 'C1-FDX is required for the assembly of mitochondrial complexes I and V in Arabidopsis' to PLOS Genetics.

The manuscript was fully evaluated at the editorial level and by three independent peer reviewers. The reviewers appreciated the attention to an important problem. While two reviewers find that the corrections made by the authors to the original submission are now adequate, the third reviewer still find some issues that require your attention, and still raised some substantial concerns about the current revised manuscript. Based on the reviews, we will not be able to accept this version of the manuscript, but we would be willing to review a revised version. We cannot, of course, promise publication at that time.

Specifically, the authors' response to the three reviews contains a wealth of information. Undoubtedly, they thoroughly addressed the concerns raised by the reviewers and made serious efforts to enhance the manuscript. However, we remain unconvinced about the data and argumentation, particularly regarding the observed effect of C1-FDX depletion on the assembly of respiratory complex V. Additionally, please note that the title, abstract and discussion parts of the paper should accurately reflect the data presented therein. Please refer to the comments provided by the reviewers (in particular to those of Rev. #2) to the authors.

If you decide to revise the manuscript for further consideration at PLOS Genetics, please aim to resubmit within the next 60 days, unless it will take extra time to address the concerns of the reviewers, in which case we would appreciate an expected resubmission date by email to plosgenetics@plos.org.

We are sorry that we cannot be more positive about your manuscript at this stage. Please do not hesitate to contact us if you have any concerns or questions.

Yours sincerely,

Oren Ostersetzer-Biran

Guest Editor

PLOS Genetics

Gregory P. Copenhaver

Section Editor

PLOS Genetics

Reviewer's Responses to Questions

**Comments to the Authors:**

Reviewer #1: Considering the revised version of the manuscript entitled “C1-FDX is required for the assembly of mitochondrial complexes I and V in Arabidopsis” by Baoyin Chen et al., I find that the authors have adequately addressed most of my comments.

Only two comments, in my previous point 4

4-The manuscript mentions that overexpressed C1-FDX-GFP leads to increased amounts of

complex I and activity. An explanation for this phenomenon should be provided.

I consider that the response was speculative and I think that the authors should reconsider the idea of increased amount of CI in the complemented plants with CI-FDX.

And in my previous point 5

5-Concerning interaction experiments, the lack of interaction of CAL2 with any other protein

raises questions. Clarification on the physiological significance of this observation is

warranted, particularly considering the interchangeability of CAL1 and CAL2 in CI models.

I find this result to be strange and believe it should be interpreted with some caution.

Reviewer #2: From reading the answers of the authors to the concerns of the reviewers I understand that the role of C1-FDX in complex I assembly is obvious but the role of C1-FDX in complex V assembly is less clear. Very last statement of the authors: „… we weakened our statement that C1-FDX is involved in the assembly of complex V in the discussion”. However, this should not only be weakened in the discussion but also in the title and in the abstract of the manuscript. Title of the revised version: „C1-FDX is required for the assembly of mitochondrial complexes I and V in Arabidopsis“. Abstract of the revised version: „Molecular analyses showed that loss of the C1-FDX function decreases the abundance and activity of both CI and CV.“ In my previous review, I pointed to the study of Röhrich et al. 2023, who also characterized two mutant Arabidopsis-lines deficient in C1-FDX. In these mutants, which seem to be true deletions, complex V abundance was not affected. I asked the authors to discuss the discrepancy of these results and the results obtained for c1-fdx-1 and c1-fdx-2. Answer of the authors: „Yes, the abundance of complex V was not observed differently between wildtype and the C1-FDX mutants in Röhrich’s data (Röhrich et al. 2023) as well as our data (Figure 2A). Because the decrease in the abundance of the α subunit is similar to the increase in the abundance of the β subunit, the loss function of C1-FDX mainly affected the assembly of F and F’.“ If I understand this statement correctly, the title of the manuscript rather should be changed into „C1-FDX is required for the assembly of mitochondrial complex I as well as subcomplexes of complex V in Arabidopsis“. Similarly, in the abstract section, it should be stated that C1-FDX absence affects the abundance of complex I and subcomplexes of complex V.

By the way, “because the decrease in the abundance of the α subunit is similar to the increase in the abundance of the β subunit, the loss function of C1-FDX mainly affected the assembly of F and F’.” – do you mean that beta-subunits might substitute alpha subunits in the holo complex V? Considering the published structures of complex V, this would be highly unlikely.

Reviewer #3: Thank you for addressing the comments and toning down your conclusions in this new version of the manuscript, although I am still not fully convinced by the conclusions about the role of C1-FDX in complex V assembly.

Just a note, BN (Blue native PAGE) is by definition non-denaturing. It uses mild detergents that isolate the protein complexes from the membranes.

**Have all data underlying the figures and results presented in the manuscript been provided?**

Reviewer #1: Yes

Reviewer #2: Yes

Reviewer #3: Yes

PLOS authors have the option to publish the peer review history of their article (what does this mean?). If published, this will include your full peer review and any attached files.

Reviewer #1: **Yes: **Eduardo Zabaleta

Reviewer #2: **Yes: **Hans-Peter Braun

Reviewer #3: No

---

## [Editor Report · Decision Letter 2]

5 Sep 2024

Dear Dr Tan,

We are pleased to inform you that your manuscript entitled "C1-FDX is required for the assembly of mitochondrial complex I and subcomplexes of complex V in Arabidopsis" has been editorially accepted for publication in PLOS Genetics. Congratulations!

Yours sincerely,

Oren Ostersetzer-Biran

Guest Editor

PLOS Genetics

Gregory P. Copenhaver

Section Editor

PLOS Genetics

Comments from the reviewers (if applicable):

**Data Deposition**

http://datadryad.org/submit?journalID=pgenetics&manu=PGENETICS-D-23-01284R2

**Press Queries**

---

## [Editor Report · Acceptance letter]

17 Sep 2024

PGENETICS-D-23-01284R2 

C1-FDX is required for the assembly of mitochondrial complex I and subcomplexes of complex V in Arabidopsis 

Dear Dr Tan, 

We are pleased to inform you that your manuscript entitled "C1-FDX is required for the assembly of mitochondrial complex I and subcomplexes of complex V in Arabidopsis" has been formally accepted for publication in PLOS Genetics! Your manuscript is now with our production department and you will be notified of the publication date in due course.

With kind regards,

Jazmin Toth

PLOS Genetics

On behalf of:
